# Syrah Grape Skin Residues Has Potential as Source of Antioxidant and Anti-Microbial Bioactive Compounds

**DOI:** 10.3390/biology10121262

**Published:** 2021-12-03

**Authors:** Roberta Barreto de Andrade, Bruna Aparecida Souza Machado, Gabriele de Abreu Barreto, Renata Quartieri Nascimento, Luiz Claudio Corrêa, Ingrid Lessa Leal, Pedro Paulo Lordelo Guimarães Tavares, Ederlan de Souza Ferreira, Marcelo Andrés Umsza-Guez

**Affiliations:** 1School of Pharmacy, Federal University of Bahia (UFBA), Salvador 40170-115, BA, Brazil or robertaba@ufba.br (R.B.d.A.); rqnutri@gmail.com (R.Q.N.); or ilessaleal@gmail.com (I.L.L.); or pedro.lordelo@ufba.br (P.P.L.G.T.); 2SENAI Institute of Innovation (ISI) in Health Advanced Systems (CIMATEC ISI SAS), University Center SENAI/CIMATEC, Salvador 41650-010, BA, Brazil; brunam@fieb.org.br (B.A.S.M.); or abreugabriele@gmail.com (G.d.A.B.); 3Brazilian Semi-Arid Agricultural Research Company (Embrapa Semiárido), BR428, Km 152, P.O. Box 23, Petrolina 56302-970, PE, Brazil; claudio.correa@embrapa.br; 4Department of Biotechnology, Health Science Institute, Federal University of Bahia (UFBA), Salvador 40170-115, BA, Brazil

**Keywords:** *Vitis vinifera*, agro-food waste, recovery, green technology, ultrasound assisted, polyphenols

## Abstract

**Simple Summary:**

The aim of this study was to verify the influence of different extraction parameters (temperature and ultrasound time) of bioactive compounds from the skin of the Syrah variety of grape. Among the extracts obtained, those exposed to 20 min of sonication had the best results in terms of flavonoid content, antioxidant potential and phenolic profile. The temperature of 60 °C provided the most relevant results for the content of total phenolics, stilbenes, flavonols and phenolic acids, however, the association of this temperature with the use of ultrasound showed lower results as a source of antioxidant and antimicrobial bioactive compounds.

**Abstract:**

In this study, we evaluated the effects of ultrasound-assisted extraction (UAE) under different time-temperature conditions on the content of bioactive compounds, antioxidant and antimicrobial activities of Syrah grape skin residue. The application of UAE showed a positive effect on the extraction of total flavonoids, and a negative effect on total polyphenols. The temperature of 40 °C and 60 °C without the UAE caused an increase of 260% and 287% of the total polyphenols, respectively. Nineteen individually bioactive compounds were quantified. The anthocyanin concentration (malvidin-3,5-di-O-glucoside 118.8–324.5 mg/100 g) showed high variation, to a lesser extent for phenolic acids, flavonoids, flavonols, procyanidins and stilbenes due to the UAE process. The Syrah grape skin residue has a high concentration of total phenolic compounds of 196–733.7 mg·GAE/100 g and a total flavonoid content of 9.8–40.0 mg·QE/100 g. The results of free radical scavenging activity (16.0–48.7 mg/100 mL, as EC_50_) and its inhibition of microbial growth (0.16 mg/mL, as EC_50_ for *S. aureus*, and 0.04 mg/mL, as EC_50_ for *E. coli*) by grape skin extract (UAE 40:20) indicate high antioxidant and antibacterial activity. It was concluded that the use of ultrasound needs further analysis for its application in this context, as it has shown deleterious effects on some compounds of interest. Syrah grape skin residue has potential as a source of bioactive antioxidants, antimicrobial activity and for use as a functional food ingredient.

## 1. Introduction

In recent years, the valorization of agro-industrial by-products has become evident, due to the extensive information available on their valuable contents of bioactive compounds that promote many beneficial effects on human health, such as reducing the risk of developing chronic disease, e.g., cancer, neurodegenerative and cardiovascular diseases [1,2].

In this sense, it includes viticulture, considered one of the most important agricultural activities around the world [3], which, from the solid winemaking by-products or grape marc, presents a high amount of residual polyphenols [4], and therefore, a low-cost source of natural antioxidants that have been used by pharmaceutical, cosmetic and food industries [5]. These phenolic compounds present in the grape by-products (pulp, stalk, seeds and peels) are known for their antioxidant, antimicrobial and anti-inflammatory properties [6,7,8].

Among these compounds, phenols stand out for their antimicrobial activity, they can penetrate the semipermeable cell membrane, and thus interact with cell proteins or cytoplasm, causing metabolic disturbances and the inhibition of microorganisms. This antimicrobial potential is greater when the phenolic acids in un-dissociated form are present in the extract composition [9,10,11].

Several authors have reported the antimicrobial activity of different grape skin extracts for different types of microorganisms, such as on Gram-positive and Gram-negative bacteria and fungi, pathogenic or not [12,13,14].

Nevertheless, the valorization of food by-products also requires additional procedures for the development of new techniques for the extraction, isolation, purification and recovery of the bioactive compounds present [15]. Overstepping these criteria, the residue can be applied as ingredients [7], improvers of technological-functional properties [16], new products [17] and for nutritional enrichment [18].

Several extraction techniques have been studied in order to optimize the recovery of existing compounds in food or by-products (seed, bagasse, skin, stalk), including ultrasound-assisted extraction (UAE) [19,20] and the use of several enzymes to degrade the vegetal matrix [21,22]. This method consists of the emission of ultrasonic waves that provide pressure variation in the liquid and, at the same time, cavitation capable of generating shear forces and microbubbles responsible for greater surface erosion, fragmentation and mass transfer [23]. The ultrasound technique is considered green technology because it requires less solvent and energy consumption, reduces the time for the process and requires moderate temperatures, which favors thermo-sensitive compounds, compared to conventional extractions, such as solid−liquid and Soxhlet [24]. Previous studies showed better results of bioactive compound extraction in grape by-products through the UAE application over conventional techniques [19,20]. Despite this, there is little information about the effect of UAE extraction using different conditions on the polyphenol compound content, and antioxidant capacity [25]. In addition, the procedures to be applied for this purpose are still not defined [26].

In this study, the effects of exposure to ultrasound to determine the optimal conditions for obtaining extracts (time and temperature) with high polyphenol content, antioxidant and antimicrobial activity from the Syrah var. grape skin were established.

## 2. Materials and Methods

### 2.1. Material and Reagents

The botanical material (grape skins) of the Syrah variety was kindly supplied by a wineries company as previously reported [8]. The skins were manually separated from the other parts and stored in freezing conditions (−20 °C) until they were used for the extraction experiments of the bioactive compounds.

The reagents used in the assays of antioxidant activity (Folin–Ciocalteau, 2,2-diphenyl-1-picrylhydrazyl [DPPH], and 2,4,6-Tris [2-pyridyl]-1,3,5-triazine [TPTZ]), total phenolics and flavonoids content (gallic acid and quercetin), and individual phenolic compounds by high performance liquid chromatography (HPLC) were purchased from Sigma-Aldrich (St. Louis, MO, USA). The solvents of HPLC grade used in the extraction (ethanol) and chromatography procedures (methanol, acetonitrile, ortho-phosphoric acid) were purchased from Merck (São Paulo, Brazil).

### 2.2. Ultrasound-Assisted Extraction (UAE)

Initially, grape skins were thawed and immediately ground fresh in a multiprocessor (Philco, Joinville, Brazil) for 10 min. Then, the procedure consisted of the extraction of bioactive compounds according to the previously established method [7], with minor adaptations. The triturated grape skin was used in the extraction process (1:5, *w*/*v*) using an ethanol solution (80%). The suspension (extract) was maintained in ultrasound (Elmasonic S30(H), Singen, Germany) to 55 ± 5 KHz, under stirring (190 rpm) at different temperatures (25 °C (Control), 40 °C, 50 °C, and 60 °C) and time (0 (Control), 10, 20, and 30 min.) conditions, as summarized in Figure 1. The suspension was then centrifuged (4800 rpm for 10 min at 25 °C) and the supernatant was used subsequently for quantification of total phenolics and flavonoids content, individual phenolic compounds, and antioxidant activity.

### 2.3. Total Phenolics and Flavonoids Content

The total phenolic content (TPC) was determined according by the Folin−Ciocalteau method [27] through spectrophotometry at absorbance of 750 nm, using a spectrophotometer (Lambda 900 UV/VIS, Perkin Elmer, Norwark, CT, USA). The standard gallic acid (10 to 200 μg/mL) was used for the calibration curve (y = 0.0104X *–* 0.0688 [R^2^ = 0.9976]). The results are presented in mg of gallic acid equivalent per 100 g of samples, on a dye weight (dw).

The total flavonoid content (TFC) was determined by the aluminum chloride colorimetric method [28] through spectrophotometry at absorbance of 415 nm, using a spectrophotometer (Lambda 900 UV/VIS, Perkin Elmer, Norwark, CT, USA). The standard quercetin (1 to 75 μg/mL) was used for the calibration curve (y = 0.0311X – 0.0259 [R^2^ = 0.9987]). The results are presented in mg of quercetin equivalent per 100 g of samples, on a dw.

### 2.4. Individual Phenolic Compounds by HPLC-DAD-FD

Nineteen bioactive compounds were quantified by HPLC (Waters 269, Alliance System, Water Corporation, Milford, MA, USA) equipped with a reverse phase column (150 mm × 4.60 mm × 3 µm, Gemini-NX C18, Phenomenex, Torrance, CA, USA), and diode array detector (DAD) and a fluorescence detector (FD). The sample of the extracts obtained under different conditions (UAE) were filtered (0.45-µm nylon) and injected in triplicate (10 µL). The gradient used was 0 min, 100% A; 10 min, 93% A; 20 min, 90% A; 30 min, 88% A; 40 min, 77% A; 45 min, 65.0% A; and 100% B at 55 min. The mobile phase systems used were (A) 0.85% phosphoric acid solution and (B) acetonitrile, with a flow rate of 0.5 mL/min., temperature from oven at 40 °C [8], and DAD conditions for the phenolic compounds, as previously established [29]. The analytical validation parameters for identification and quantification of phenolic compounds were determined, as follows: standard calibration curve (concentration range of 0.3125–40 µg/mL), regression coefficients equations ([R^2^] ranged from 0.9973–0.9996), theoretical limits of detection (0.004–0.1760 μg/mL), and quantification (LOQ) (0.0130–0.5870 μg/mL), average recovery value (anthocyanins 92.70 to 116.39%; flavonols 88.92 to 104.94%; phenolic acids 82.27 to 93.91% and tannins 97.10 to 112.42%), and accuracy of the RSD method (0.73–2.00% for nonenriched samples and 0.71–3.22% for enriched samples were determined. The results are presented in mg/100 g, on a dw.

### 2.5. Antioxidant Activity

DPPH free radical scavenging was performed according to the established method [30]. After adding the extracts obtained by ultrasound-assisted extraction (Figure 1), the decrease in absorbance at 517 nm of DPPH 100 mM, it was measured after 30 min byn spectrophotometer (Lambda 900 UV/VIS, Perkin Elmer, Norwark, CT, USA). The free radical scavenging was expressed in EC_50_ µg per g of samples (EC_50_ means the antioxidant concentration required to obtain a 50% radical inhibition), on a dw.

### 2.6. Antimicrobial Activity

The minimum inhibitory (MIC) and minimum bactericidal (MBC) concentration, and the minimum fungicidal concentration (MFC) were performed according to the reference standard procedures (CLSI M 38-A, 2002). Briefly, the microorganisms with a final cellular density of 1.5 × 105 CFU/mL, were used in microplates incubated in a bacteriological incubator for 72 h at 28 °C. The Syrah var. grape skin extracts were tested from a serial dilution (10 mg/mL to 0.02 mg/mL). Amphotericin B diluted in dimethylsulfoxide (DMSO) was used as a positive control (16 e 0.03 µg/mL), and DMSO absent of extracts or amphotericin B as a negative control. 

In the antibactericidal activity assays were used the *Staphylococcus aureus* ATCC 25,923 and *Escherichia coli* ATCC 25,922 strains. The MIC was performed according to the previously established procedure [31] with minor modifications. Briefly, an aliquot of 100 µL of the grape skin extracts, previously solubilized in 10% (*w*/*v*) ethanol and Milli-Q water at different concentrations (2.5 mg/mL to 0.001 mg/mL) was used. The plates were incubated for 24 h at 37 °C and the MIC was defined as the lowest concentration of extract capable of preventing microbial growth.

### 2.7. Statistical Analysis

All results were expressed as mean ± standard deviation (SD) for at least three independent analyses. One-way analysis of variance (ANOVA), and the Turkey test (Statistica, v. 7.0, StatSoft software) was applied to evaluate the difference between means (*p* ≤ 0.05). The covariance relationships between the content of phenolic compounds and the free radical scavenging of the extracts obtained by UAE were evaluated by Pearson’s correlation coefficient (r).

## 3. Results and Discussion

### 3.1. Bioactive Compounds by UAE from Syrah Var. Grape Skin Residue

In the literature, different studies that aim at higher yields of phenolic compounds from plant matrices correlate the oscillation of results to the application of process variables, such as use time and temperature variation [32,33,34]. Data regarding the content of total phenolic compounds are often associated with the capacity of antioxidant properties [5,35].

Figure 2 shows the results obtained for both parameters (time and temperature) together with the total content of phenolic compounds and flavonoids in each extract. It was verified that the extracts obtained at a temperature of 60 °C with and without sonication had a higher concentration of phenolic compounds, ranging between 543 ± 14 and 759 ± 27 mg·GAE/100 g. These results are statistically superior to those obtained by extracting bioactive compounds at room temperature without the application of ultrasound (25 °C control). Among the extracts that were exposed to ultrasound, the UAE 60:30 extract showed the greatest increase in the content of phenolic compounds compared to the control at 25 °C (73.3%). This increase occurs because the polysaccharides that make up the cell wall of the grape skin, at high temperatures, undergo hydrolysis, releasing a greater concentration of phenolic compounds [36,37].

Combined with temperature, ultrasound also contributes to the breakdown of grape skin cells [25], allowing the penetration of the solvent into the sample and aiding in the process of extracting the compounds of interest [38]. However, when comparing the content of phenolic compounds in the extracts obtained by ultrasound with their respective controls, it is noted that these had lower values.

The increase in temperature in the extraction processes positively affected the content of flavonoids and phenolic compounds in the extracts, especially those obtained at 60 °C with ultrasound. Rodríguez-Pérez, Quirantes-Piné, Fernández-Gutiérrez, and Segura-Carretero [39] found that the use of ultrasound for 15 min provided extracts with a predominance of flavonoids among the phenolic compounds extracted. Vodnar et al. [31] found results lower than those reported in this research for the flavonoid content, however, this can be explained by the difference in the phenolic composition of the grapes, due to the interference of external agents of cultivation.

From the results, it was noted that the application of ultrasound favored the extraction of flavonoids, but impaired the content of phenolic compounds in the extract. The use of ultrasound also contributed to better results in terms of antioxidant activity, and this can be explained by the greater extraction of flavonoids from the grape skin, as reported in other studies [25,40].

### 3.2. Individual Bioactive Compounds by UAE from Syrah Var. Grape Skin Residue

The extracts obtained were analyzed by HPLC, and 19 individual phenolic compounds were quantified (Table 1). The best extracts were chosen in terms of total flavonoids at different temperatures (40, 50 and 60 °C) and ultrasound for 20 min from the different processes. The extract prepared at room temperature (25 °C) applying 20 min of ultrasound (UAE 25:20) was prepared to verify whether the ultrasound applied in the absence of high temperatures was capable of enhancing the extraction or degrading of compounds.

In the group of phenolic acids, caftaric acid was the one with the highest concentration in all extracts, except in UEA 60:20. Studies report caffeic acid as the main phenolic acid [8,41,42,43] in grape skins. Caffeic acid was the only compound of the group present in all extracts, however, the values verified were lower than those registered in the whole grape residue extracts [44]. In grape juices, gallic acid and phenolic acid are quantified in higher concentrations, however, this compound is mainly found in grape seeds [29].

In UAE 50:20 and UAE 60:20 extracts, trans-resveratrol was not determined in the analyses. According to Casazza, Aliakbarian, Mantegna, Cravotto, and Perego [45], trans-resveratrol is found in greater amounts in grape seeds than in skins. In the work of Natividade et al. [29], it was found that the content of trans-resveratrol present in the Syrah variety grape juice was also below the detection limit. The high temperatures associated with the use of ultrasound showed negative effects on the individual flavonoids in the extracts, since the control at 60 °C showed higher values for all compounds, compared to the US 60:20 extract. The same could be verified with the use of ultrasound at room temperature. Teixeira et al. [1] reported that procyanidin B2 is one of the most abundant flavonoids present in grape skins and seeds, a fact observed in this study, as this compound from the group of flavanols was the one with the highest concentration in the extracts.

Sonication did not influence the extraction of compounds from the flavonol group. The concentration of myricetin, for example, was higher in all extracts prepared without ultrasound pretreatment. Regarding the content of quercetin-3-β-D-glycoside and isohramnetin-3-O-glycoside in the samples, only the UAE 50:20 extract presented values higher than the control (50 °C). The compounds kaempferol-3-O-glucoside and rutin showed results with opposite trends: the use of ultrasound was relevant only for the former at room temperature, while the latter had higher concentrations in all extracts with the application of ultrasound, except at 60 °C. In general, these results are not in agreement with the work by Caldas et al. [33], in which the authors found greater extraction of flavonols with the use of ultrasound.

Among the anthocyanins, the compounds with the highest values were peonidin-3-O-glucoside, pelargonidin-3-O-glucoside and malvidin-3-O-glucoside. Melo et al. [46], when evaluating the phenolic profile of Syrah grape pomace grown in the region of the São Francisco Valley, found that the levels of peonidin-3-O-glycoside and malvidin-3-glycoside were also predominant in the anthocyanin group.

In this study, the malvidin-3-O-glucoside concentration range found for the extracts (118.8 ± 5.2 to 324.5 ± 3.2 mg/100 g) by UAE was greater than the values found in red grape pomace by conventional extraction, ultrasound-assisted extraction and microwave-assisted extraction [33].

It is important to emphasize that the amount of anthocyanins present in the grape is influenced by cultivation factors, such as: grape planting method, climatic aspects, physicochemical parameters (pH and temperature) [47,48].

The sum of the concentrations of the phenolic compounds in each group, under different extraction conditions are shown in Figure 3. It can be observed that the chemical groups with the highest concentrations in the extracts were anthocyanins and flavonols. Lingua, Fabani, Wunderlin, and Baroni [49] found that, similarly, the two groups were the main ones found in three different grape varieties (Syrah, Merlot, Cabernet Sauvignon), with anthocyanins being the compounds with the highest concentration in Syrah grapes. The authors also highlighted that malvidin and quercetin were the compounds in the highest concentrations in the samples, as shown in Table 1.

The application of high temperatures for the preparation of extracts showed a negative influence on the concentration of anthocyanins. The control sample at 25 °C, without ultrasound exposure, had the highest anthocyanin content, followed by the UAE 25:20 sample, both with statistically different values from the other extracts. With the exception of extracts prepared at a temperature of 50 °C, the use of ultrasound negatively influenced the anthocyanin content. Tiwari, Patras, Brunton, Cullen, and O’Donnell [50] reported possible changes in the concentrations of anthocyanins delphinidin-3-O-glucoside, cyanidin-3-O-glucoside and malvidin-3-O-glucoside, when the ultrasound was applied under different conditions (time of exposure to the ultrasound and amplitude level used in the equipment). Some authors suggest that the degradation of anthocyanins may be related to oxidation reactions caused by the interaction with free radicals formed during extraction using ultrasound [51]. The group of flavanols and flavonols showed similar behavior to that of anthocyanins, regarding the use of ultrasound, that is, all extracts exposed for 20 min to ultrasound reached values lower than those verified in extracts subjected exclusively to the influence of temperature, with the exception of the extracts prepared at 50 °C. These results suggest a possible disadvantage to the use of ultrasound, due to the degradation of the compounds.

As noted in this study, Li et al. [52] reported that flavonoids, whose main compounds are anthocyanins, flavanols and flavonols, are in greater abundance in grape skins. This characteristic is associated with the maturation of the grapes used in winemaking: the more ripe, the greater the occurrence of anthocyanins in the extracts and the lower the anthocyanin/flavanol ratio [53]. With the exception of flavonoids, phenolic acids are compounds present in plant matrices with a relevant biological effect [54].

In the analyzed samples, the phenolic acid concentration range (27.6 to 46.9 mg/100 g) was the lowest among the phenolic groups. This is because this class of phenolic compounds is found in greater concentration in grape pulp [52].

Stilbenes, a group that includes the trans-resveratrol compound, had the lowest concentration in the samples, however, fresh grape skins are indicated as an excellent source of resveratrol, with concentrations ranging from 50 to 100 µg/g [55]. Rockenbach et al. [56] did not detect the presence of trans-resveratrol in the skin of wine residue grapes. The authors suggest that the compound was transferred to the product made from grapes during manufacturing, which also justifies the low concentration of stilbenes in the extracts in this study. Fernández-Marín et al. [57] stated that the terroir is decisive for the stilbene content present in the grapes.

### 3.3. Free Radical Scavenging Activity of Extracts Obtained from Syrah Var. Using UAE

The content of phenolic compounds is generally associated with the presence of antioxidant activity in different plant matrices, including grapes [24].

To assess the antioxidant potential, the EC50 method was used, which calculates the amount of antioxidants needed to inhibit the initial concentration of the DPPH radical by 50% [58], therefore, the lower the concentration value, the higher will be the antioxidant capacity (Figure 4a). All extracts exposed for 20 min to ultrasound (UAE 40:20; UAE 50:20; UAE 60:20) are included in the group with the highest antioxidant potential (22, 20 and 16 mg/100 mL, respectively). These samples, when related to controls at each temperature, showed the increase of antioxidant activities in 47.6%, 50.0% and 61.0%, respectively. EC50 values above 250 mg/mL refer to the low antioxidant potential of the sample [59], therefore, it is possible to state that all extraction conditions applied in this research contributed to obtaining extracts with excellent antioxidant capacity, since the values ranged from 16.8 ± 0.2 to 48.7 ± 0.5 mg/100 mL. 

Some research carried out with grape skin extracts demonstrate the anti-radical potential, such as, Sridhara and Charles [60] who evaluated the scavenging activity of DPPH of grape skin extracts (var. Kyoho) and verified values between 77.92 and 95.71 mg/mL at the different concentrations tested. Park et al. [61] observed in three extracts (peel/pulp) of different grape varieties scavenging activity between 48.47 and 57.09%. Doshi et al. [62] verified antiradical activity of 4.2 and 1.8 (TE mM/mL) Pusa Navrang and Merlot variety, respectively.

From the values obtained from some compounds, the possible correlation between them was analyzed. The results for this assessment are shown in Table 2. The correlation between the total anthocyanin group and the compound malvidin-3-glucosside-chloride reached a value close to 1, thus being a positive correlation. This result ensures that the greater the total anthocyanin content in the sample, the higher the concentration of malvidin-3-glucoside-chloride, as predicted from previous results. Similar to this study, Antoniolli et al. [63] found that malvidin-3-glucoside-chloride was the predominant compound among the total anthocyanins analyzed in grape pomace.

Both total anthocyanins and malvidin-3-glucoside-chloride did not correlate with the antioxidant activity of the extracts. According to Orak [64], the absence of correlation had already been confirmed in other studies. The total flavonoid content was the only one that achieved a significant correlation in terms of antioxidant potential (*p* < 0.0001). The negative value close to 1 suggests that these parameters behave in an inversely proportional manner. Knowing that the lower the value for IC_50_, the greater the antioxidant capacity, it is noted that the content of flavonoids has a positive influence for this parameter. The correlation between the total phenolic content and the total flavonoid content showed a value closer to zero than 1 (0.441), suggesting the lack of correlation between these parameters. The same occurred with the correlation between total phenolic content and procyanidin B2 (0.451).

Quercetin was also applied in the correlation analyses; however, it is not registered in Table 2 because it did not present statistically significant results with at least one of the parameters (*p* < 0.05). This fact, as well as the diffuse values for correlation, raise the possibility of degradation of some bioactive compounds due to the use of ultrasound during extraction [24].

### 3.4. Antimicrobial Activities of Extracts Obtained from Syrah Var. Using UAE

Some researchers highlight the antimicrobial action of compounds (such as polyphenols, flavonoids, others) preferentially present in grape skins [9,10] that could be used as natural preservatives in food [8]. As examples of these compounds, stilbenes and resveratrol can be cited, which act against phytopathogens such as Botrytis cinerea [12]. Resveratrol also has recognized antimicrobial activity against several species, such as *Trichophyton tonsurans, Staphylococcus aureus, Pseudomonas aeruginosa, Enterococcus faecalis, Epidermophyton floccosum, Microsporum gypseum* and *Trichophyton mentagrophytes*.

The extracts UAE 40:20, UAE 50:20, UAE 60:20 and their respective controls (40 °C, 50 °C and 60 °C) and the extract prepared without the application of ultrasound at room temperature (C:A) were also subjected to this analysis. However, the antibacterial activity was only observed in the UAE 40:20 extract (Figure 4b). This situation needs further research because the concentrations of phenolic acids, stilbenes, flavonols, flavanols, anthocyanin compounds, in addition to the free radical scavenging activity were almost the same in the evaluated samples. Many scientific works report that grape seeds contain greater amounts of total phenolic compounds when compared to skins, thus, extracts obtained from seeds have greater antimicrobial activities [61].

Figure 4b shows that, for *E. coli*, the extract showed high antibacterial activity up to a concentration of 0.62 mg/mL, with an MIC of 1.25 mg/mL and an MBC of 2.5 mg/mL. *Staphylococcus aureus*, on the other hand, was less sensitive to the antibacterial action of the UAE 40:20 extract, with MIC and MBC values at a concentration of 2.5 mg/mL. Contrary to what was shown in this study, Yeo, Leo, and Chan [65] and Silva et al. [8] found that the propolis extract obtained from UAE had better effects against Gram-positive bacteria than Gram-negative ones. Oliveira et al. [66] in a study carried out with grape pomace extract (var. Merlot and Syrah) obtained by supercritical extraction (SFE), reported that Gram-negative bacteria were more resistant to the antimicrobial effects of the extract, when compared to Gram-positive bacteria, however, the authors revealed that, for the extracts obtained by SFE with the application of ethanol as a co-solvent, no antimicrobial activity was observed. Katalinić et al. [13] used extracts made from the skins of 14 different grape varieties as an antimicrobial agent and showed that there were no significant differences in the resistance of Gram-positive and Gram-negative bacteria. Katalinić et al. [13] verified the antimicrobial effect (Gram-positive bacteria: *Staphylococcus aureus, Bacillus cereus* and Gram-negative bacteria *Escherichia coli* O157:H7, *Salmonella Infantis*, *Campylobacter coli*) of extracts from 14 grape skin varieties using the technique of microdilution. The minimum inhibitory concentrations (0.014–0.59 mg·GAE/mL), were found especially against Campylobacter and Salmonella.

Gram-negative bacteria have two layers in their cell membrane, unlike Gram-positive bacteria that have only one barrier, facilitating the penetration of lipophilic compounds [67,68]. However, according to Yeo et al. [65], the physicochemical characteristic of the extracts is the main factor for the distinct bioactive qualities between the samples, followed by the structural difference of the bacteria. The grape extracts under the conditions and concentration ranges tested were not able to inhibit the growth of the isolates (data not shown). The MIC and CFM for amphotericin B were 0.5 µg/mL and 4 µg/mL, respectively, for the *Aspergillus fumigatus* isolate, and 4 µg/mL and 16 µg/mL, for the *Fusarium oxysporum sp. passiflorae*.

## 4. Conclusions

In this study, it was found that within the extracts made with ultrasound technology, those exposed to 20 min of sonification achieved the best results in terms of flavonoid content, antioxidant potential and phenolic profile. The temperature of 60 °C provided the most relevant results for the content of total phenolics, stilbenes, flavanols and phenolic acids, however, the association of this temperature with the use of ultrasound showed lower results. In this sense, the use of ultrasound, despite being inserted in new technologies, requires more analysis for its application in this context, since it has been shown to have deleterious effects in some compounds of interest. Finally, the Syrah grape skin residue presents potential as a source of antioxidant and antimicrobial bioactive compounds.

## Figures and Tables

**Figure 1 biology-10-01262-f001:**
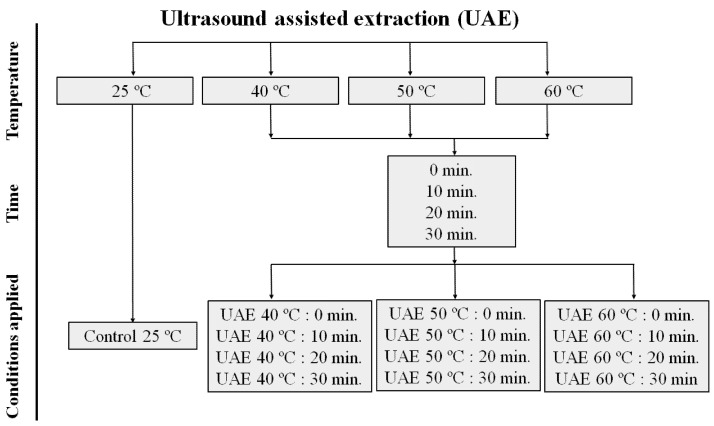
Production conditions of the Syrah var. grape skin residue extracts. Ultrasound-assisted extraction (UAE) at temperature 40 °C for 0 (UAE 40 °C: 0 min), 10 (UAE 40 °C: 10 min), 20 (UAE 40 °C: 20 min) and 30 (UAE 40 °C: 30 min), 50 °C for 0 (UAE 50 °C: 0 min), 10 (UAE 50 °C: 10 min), 20 (UAE 50 °C: 20 min) and 30 (UAE 50 °C: 30 min), and 60 °C for 0 (UAE 60 °C: 0 min), 10 (UAE 60 °C: 10 min), 20 (UAE 60 °C: 20 min), and 30 (UAE 60 °C: 30 min), respectively.

**Figure 2 biology-10-01262-f002:**
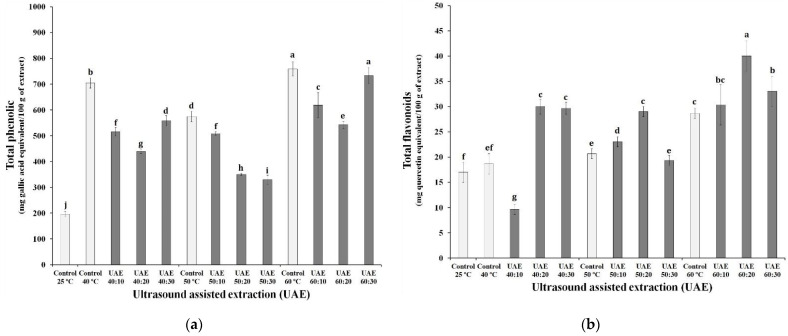
Effect of the ultrasound-assisted extraction using (**a**) different time-temperature processes in the total phenolics, and (**b**) total flavonoids of the Syrah var. grape skin residue. Ultrasound-assisted extraction (UAE) at temperature 40 °C for 0 (UAE 40 °C: 0 min), 10 (UAE 40 °C: 10 min), 20 (UAE 40 °C: 20 min) and 30 (UAE 40 °C: 30 min), 50 °C for 0 (UAE 50 °C: 0 min), 10 (UAE 50 °C: 10 min), 20 (UAE 50 °C: 20 min) and 30 (UAE 50 °C: 30 min),; and 60 °C for 0 (UAE 60 °C: 0 min), 10 (UAE 60 °C: 10 min), 20 (UAE 60 °C: 20 min) and 30 (UAE 60 °C: 30 min), respectively. Values with different superscript letters (a–h) between treatments differ significantly (*p* < 0.05) by the Turkey test multiple range tests. Each value is the mean ± SE (n = 3).

**Figure 3 biology-10-01262-f003:**
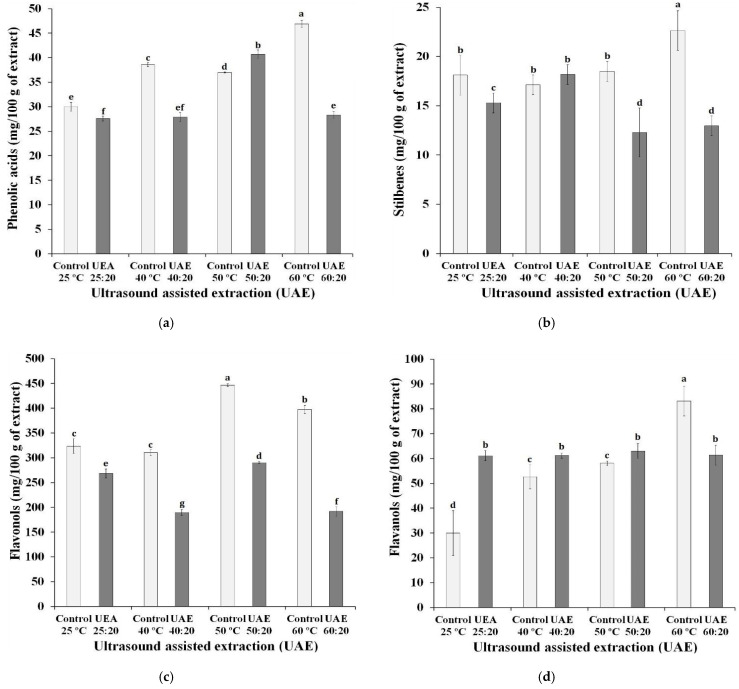
Effect of ultrasound-assisted extraction using different time-temperature processes on the content of (**a**) phenolic acids, (**b**) stilbenes, (**c**) flavonols, (**d**), flavanols and (**e**) anthocyanin compounds in Syrah var. Ultrasound-assisted extraction (UAE) at temperature 40 °C for 0 (UAE 40 °C: 0 min), 10 (UAE 40 °C: 10 min), 20 (UAE 40 °C: 20 min) and 30 (UAE 40 °C: 30 min), 50 °C for 0 (UAE 50 °C: 0 min), 10 (UAE 50 °C: 10 min), 20 (UAE 50 °C: 20 min) and 30 (UAE 50 °C: 30 min), and 60 °C for 0 (UAE 60 °C: 0 min), 10 (UAE 60 °C: 10 min), 20 (UAE 60 °C: 20 min) and 30 (UAE 60 °C: 30 min), respectively. Values with different superscript letters (a–g) between treatments differ significantly (*p* < 0.05) by the Turkey test multiple range tests. Each value is the mean ± SE (n = 3).

**Figure 4 biology-10-01262-f004:**
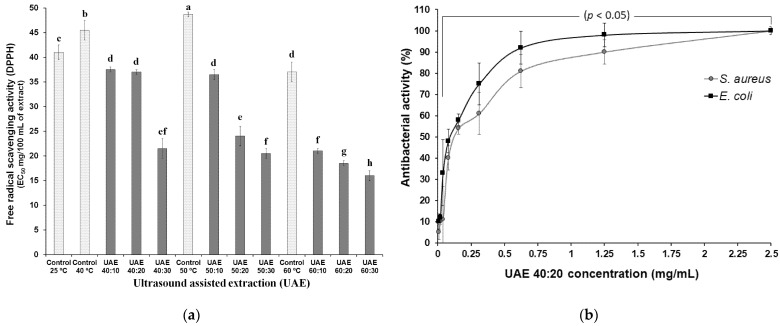
Effect of ultrasound-assisted extraction using different processes (time-temperature) (**a**) on the free radical scavenging activity and (**b**) on the antibacterial activity of the extract of skin of grape var. Syrah. Ultrasound-assisted extraction (UAE) at temperature 40 °C for 0 (UAE 40 °C: 0 min), 10 (UAE 40 °C: 10 min), 20 (UAE 40 °C: 20 min) and 30 (UAE 40 °C: 30 min), 50 °C for 0 (UAE 50 °C: 0 min), 10 (UAE 50 °C: 10 min), 20 (UAE 50 °C: 20 min) and 30 (UAE 50 °C: 30 min), and 60 °C for 0 (UAE 60 °C: 0 min), 10 (UAE 60 °C: 10 min), 20 (UAE 60 °C: 20 min) and 30 (UAE 60 °C: 30 min), respectively. Values with different superscript letters (a–h) between treatments differ significantly (*p* < 0.05) by the Turkey test multiple range tests. Each value is the mean ± SE (n = 3).

**Table 1 biology-10-01262-t001:** Effect of ultrasound-assisted extraction under different process conditions (time-temperature) on the quantification of individual compounds present in Syrah var.

Phenolic Compounds mg/100 g	Extracts ^‡^							
	Control 25 °C	UAE 25:20	Control 40 °C	UAE 40:20	Control 50 °C	UAE 50:20	Control 60 °C	UAE 60:20
**Phenolic acids**								
Gallic acid	9.6 ± 0.1 ^b^	<LQ	9.3 ± 0.5 ^b^	<LQ	10.2 ± 0.0 ^ab^	<LQ	14.9 ± 0.3 ^a^	<LQ
Caffeic acid	7.3 ± 0.1 ^ab^	6.4 ± 0.1 ^b^	8.6 ± 0.1 ^ab^	6.8 ± 0.0 ^ab^	8.4 ± 0.0 ^ab^	8.2 ± 0.1 ^ab^	9.9 ± 0.2 ^a^	7.9 ± 0.2 ^ab^
Caftaric acid	17 ± 0.1 ^ab^	14.2 ± 0.4 ^b^	20.7 ± 0.3 ^ab^	13.9 ± 0.1 ^b^	18.4 ± 0.1 ^ab^	22.4 ± 0.5 ^a^	22.1 ± 0.2 ^a^	<LQ
**Stilbenes**								
Cis-resveratrol	3.5 ± 0.1 ^ab^	2.9 ± 0.0 ^b^	2.9 ± 0.0 ^b^	3.6 ± 0.0 ^ab^	3.2 ± 0.0 ^b^	ND	4.3 ± 0.0 ^a^	3.6 ± 0.0 ^ab^
Trans-resveratrol	5.5 ± 0.1 ^ab^	4.7 ± 0.0 ^b^	4.9 ± 0.2 ^b^	5.0 ± 0.0 ^b^	5.8 ± 0.0 ^ab^	ND	6.3 ± 0.0 ^a^	ND
Viniferin	9.1 ± 0.2 ^bc^	7.6 ± 0.0 ^c^	9.3 ± 0.3 ^bc^	9.5 ± 0.0 ^abc^	9.5 ± 0.0 ^abc^	10.4 ± 0.1 ^ab^	12.1 ± 0.2 ^a^	9.4 ± 0.0 ^bc^
**Flavanols**								
(+)-Catechin	14.9 ± 0.5 ^ab^	12.9 ± 0.2 ^b^	16.5 ± 0.4 ^ab^	13.7 ± 0.0 ^b^	16.0 ± 0.3 ^ab^	14.6 ± 0.4 ^ab^	23.0 ± 0.4 ^a^	14.8 ± 0.2 ^ab^
Procyanidin B1	21.1 ± 0.4 ^ab^	18.4 ± 0.1 ^b^	20.2 ± 0.8 ^b^	22.5 ± 0.1 ^ab^	19.7 ± 0.4 ^b^	23.8 ± 0.2 ^ab^	27.1 ± 0.2 ^a^	22.5 ± 0.2 ^ab^
Procyanidin B2	25.1 ± 0.5 ^ab^	21.4 ± 0.2 ^b^	24.4 ± 0.5 ^ab^	23.7 ± 0.1 ^b^	22.4 ± 0.3 ^b^	24.6 ± 0.4 ^ab^	33.0 ± 0.4 ^a^	24.0 ± 0.3 ^b^
**Flavonols**								
Kaempferol-3-O-glucoside	12.1 ± 0.3 ^a^	13.4 ± 0.4 ^a^	12.5 ± 0.2 ^a^	10.4 ± 0.0 ^a^	16.8 ± 0.1 ^a^	6.7 ± 0.4 ^a^	19.9 ± 0.5 ^a^	11.1 ± 0.3 ^a^
Quercetin-β-D-glucoside	199.1 ± 5.1 ^c^	177.4 ± 3.8 ^de^	187.7 ± 6.2 ^cd^	110.4 ± 4.4 ^f^	167.9 ± 2.8 ^e^	288.9 ± 3.2 ^a^	247.7 ± 2.7 ^b^	112.5 ± 0.4 ^f^
Isorhamnetin-3-glucoside-chloride	68.4 ± 0.3 ^c^	40.1 ± 0.3 ^d^	65.3 ± 0.2 ^c^	39.3 ± 0.2 ^d^	64.5 ± 0.1 ^c^	99.0 ± 0.1 ^b^	77.6 ± 0.1 ^b^	43.8 ± 0.2 ^d^
Myricetin	37.4 ± 0.4 ^abc^	29.8 ± 0.5 ^c^	39.6 ± 0.2 ^ab^	17.7 ± 0.1 ^d^	34.1 ± 0.2 ^bc^	33.5 ± 0.3 ^bc^	43.9 ± 0.3 ^a^	16.2 ± 0.3 ^d^
Rutin	6.7 ± 0.2 ^ab^	7.4 ± 0.1 ^ab^	5.1 ± 0.1 ^b^	7.6 ± 0.0 ^ab^	6.9 ± 0.1 ^ab^	8.4 ± 0.1 ^ab^	8.7 ± 0.1 ^a^	8.4 ± 0.2 ^ab^
**Anthocyanins**								
Malvidin-3-glucoside-chloride	324.5 ± 3.2 ^a^	259.8 ± 1.5 ^b^	150.0 ± 6.4 ^d^	118.8 ± 5.2 ^e^	121.6 ± 2.9 ^e^	157.5 ± 3.3 ^d^	203.3 ± 3.8 ^c^	159.8 ± 5.3 ^d^
Cyanidin-3-glucoside-chloride	5.5 ± 0.1 ^ab^	4.8 ± 0.0 ^bc^	5.1 ± 0.2 ^bc^	5.8 ± 0.0 ^ab^	4.3 ± 0.1 ^c^	6.4 ± 0.0 ^a^	6.9 ± 0.0 ^a^	ND
Pelargonidin-3-glucoside-chloride	41.2 ± 0.5 ^a^	33.8 ± 0.5 ^ab^	21.9 ± 0.2 ^cd^	19.6 ± 0.1 ^d^	18.6 ± 0.1 ^d^	23.9 ± 0.1 ^cd^	28.0 ± 0.1 ^bc^	25.5 ± 0.1 ^cd^
Delfinidine-3-O-glucoside	17.4 ± 0.3 ^a^	12.4 ± 0.4 ^abc^	10.9 ± 0.2 ^bc^	11.0 ± 0.1 ^abc^	9.3 ± 0.1 ^c^	10.3 ± 0.2 ^bc^	16.0 ± 0.2 ^ab^	11.5 ± 0.2 ^abc^
Peonidine-3-O-glucoside	34.2 ± 0.5 ^a^	31.1 ± 0.2 ^a^	13.8 ± 0.3 ^d^	22.0 ± 0.1 ^bc^	14.7 ± 0.1 ^d^	24.6 ± 0.2 ^bc^	28.1 ± 0.2 ^ab^	19.5 ± 0.1 ^cd^

^‡^ The values (means ± SD) correspond to averages from three replicates. Different letters in the same line indicate significant differences between the values (*p* < 0.05). <LQ = limit of quantification; ND = not detected. Ultrasound-assisted extraction (UAE) at temperature 40 °C for 0 (UAE 40 °C: 0 min), 10 (UAE 40 °C: 10 min), 20 (UAE 40 °C: 20 min) and 30 (UAE 40 °C: 30 min), 50 °C for 0 (UAE 50 °C: 0 min), 10 (UAE 50 °C: 10 min), 20 (UAE 50 °C: 20 min) and 30 (UAE 50 °C: 30 min), and 60 °C for 0 (UAE 60 °C: 0 min), 10 (UAE 60 °C: 10 min), 20 (UAE 60 °C: 20 min) and 30 (UAE 60 °C: 30 min), respectively.

**Table 2 biology-10-01262-t002:** Correlation between phenolic compounds and antioxidant activity.

Variables	EC50	Total Flavonoid Content	Malvidin-3-Glucoside-Chloride	Total Anthocyanins	Total Phenolic Content	Procyanidin B2
EC50	-	−0.737 (<0.0001)	0.480 (0.018)	0.454 (0.026)	-	-
Total flavonoid content	−0.737 (<0.0001)	-	−0.478 (0.018)	−0.457 (0.025)	0.441 (0.031)	-
Malvidin-3-glucoside-chloride	0.480 (0.018)	−0.478 (0.018)	-	0.996 (<0.0001)	−0.657 (0.000)	-
Anthocyanins	0.454 (0.026)	−0.457 (0.025)	0.996 (<0.0001)	-	−0.649 (0.001)	-
Total phenolic content	-	0.441 (0.031)	−0.657 (0.000)	−0.649 (0.001)	-	0.451 (0.027)
Procyanidin B2	-	-	-	-	0.451 (0.027)	-

## Data Availability

Not applicable.

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
