# Peer review of "Syrah Grape Skin Residues Has Potential as Source of Antioxidant and Anti-Microbial Bioactive Compounds"

_biology, 2021, doi:10.3390/biology10121262_

Round 1

Reviewer 1 Report

The manuscript by Roberta Barreto de Andrade and co-workers presents the results on the optimization of UAE extraction of polyphenolic compounds from the skins of Syrah grape, with antioxidant and anti-bacterial activity.

The topic of using food industry by-products as a source of bioactive compounds for pharmaceutical, cosmetic or food applications is very important due to the global problems with waste management and declining areas suitable for the cultivation of medicinal plants. However, I am not convinced if the manuscript regarding this topic is suitable for Biology journal - in my opinion there are other MDPI journals with more appropriate aim and scope for this type of scientific topic (eg. Applied Sciences or Antioxidants).

I have also a few suggestions regarding the content of the manuscript, as listed below:

The Introduction is too general – there is a lot of literature data regarding the extraction methods used to obtain active compounds from grape by-products and the Authors should refer to these data in order to underline the originality of their work. It would also be important to present Syrah grape variety as a source of bioactive compounds and refer to other Authors extracting photochemicals from the by-products derived from this variety (eg. Meini MR, Cabezudo I, Boschetti CE, Romanini D. Recovery of phenolic antioxidants from Syrah grape pomace through the optimization of an enzymatic extraction process. Food Chem. 2019 Jun 15;283:257-264.)

In Materials and Methods section please specify:

- which type of alcohol solution was used for extraction (line 98)-

- if the plant material used for extraction was frozen, thawed, dried (line 99) – the Authors indicated in line 84 that the fresh material was frozen until analysis but it was not explained how it was prepared for the extraction
- Figure 1 – please add the units for the values 25, 40, 50, 60 and 10, 20, 30
- please consider preparation of an additional table summarizing prepared extracts and their abbreviations as it would make it easier to follow the results
- in the description of the antioxidant activity method (lines 151-157) please indicate what was the negative control and if any positive control/reference antioxidant was used for comparison; please explain the abbreviation EC50
- line 161 – it would be more accurate to change “cell concentration” to “cellular density”
- line 162 – it would be more accurate to change “oven” for “incubator”
- line 164 – please change “Amphotericin B diluted in DMSO was used as a positive control”

Results and Discussion:
- The data presented in Figure 2 indicate the content of phenolic compounds and flavonoids in 13 extracts, however the analysis of individual compounds summarized in Table 1 was performed only for 8 extracts. Could you please explain why not all the extracts were subjected to HPLC analysis?
- please correct the values in line 328 (d6%, 50%, 6th)
- in line 329 the Authors are referring to IC50 values, however Figure 4 (a) shows EC50 – correct
- in order to compare the antioxidant data obtained by the Authors with other studies on grape by-products extracts it would be convenient to add the data on IC50 values obtain by others and/or include the results for a known antioxidant in Figure a (a)
- in my opinion it would be more clear to describe the correlation between the phenolic content and antioxidant activity just after the description of the antioxidant activity of the extracts and move the antimicrobial studies to a separate chapter
- the discussion of the results indicating antimicrobial activity of Syrah grape extracts should be revised – please describe in more details other antimicrobial studies performed using grape skin/pomace extracts, the names of the bacterial strains used in other studies should be added, as well as the MIC values; are there any information regarding the compounds isolated from grapes that are responsible for observed antibacterial activity?

Author Response

Please find enclosed the revised version of the manuscript entitled “Syrah grape skin residues has potential as source of antioxidant and anti-microbial bioactive compounds”. We would like to thank the reviewers for their criticisms and considerations, which contributed positively to the scientific quality of the manuscript. All comments and issues raised by the referee were dealt with and resulted in the modifications described below and highlighted in the manuscript

Reviewer 2 Report

This manuscript has evaluated the effects of UAE under different time-temeprature conditions in order to evaluete the biocative compounds of Syra grape skin residue. Although the approach is very interesting, there are some aspects which are, according to my point of view, not clear and should need to be clarified:

Introduction: I would recommend extending the introduction, including some previous examples in the antioxidant and antimicrobial activity area in this context
Materials and Methods 
2.2. Due to the fact that the whole article is based on the naming of the samples based on their extraction conditions, the different extraction conditions would need to be explained in more detail, as it is not clear enough. A table may help understand better the naming of the samples.

Figure 1. To make it clearer, I would add min. to each time. Ex.: 25 min, 40 min

Results and Discussion
L266-267: I cannot find the malvidin values in Table 1.
L338-339: "...The antibacterial activity was only observed in the UAE 40:20 extract" Why? A more in depth discussion of this topic would be advisable.

Minor points:

L49: ...The valorization any a food by-products... This sentence would be rewrite
L 131: The word Caffeine is duplicated
L297: The reference has all letters in capital letters
L328: There is a mistake d6%

Author Response

(The authors gave the same response as above.)

Reviewer 3 Report

In this study, authors evaluated the effects of ultrasound-assisted extraction on the content of bioactive compounds, antioxidant and antimicrobial activities of Syrah grape skin residue. The paper is well written, but minor revision before publishing (listed in later text) is needed. All technical errors need to be corrected.

  1. Cell concentration (line 161)
  2. Mass concentration of amphotericin B (line 164)
  3. of has been entered twice (line 295)
  4. Please correct the increase of antioxidant activities (line 328)
  5. Figure 4b. The values of the mass concentration on the x-axis should be directed from lower to higher concentrations

this manuscript do not represent a significant uppgrade in the study of ultrasound-assisted extraction of bioactive compounds, antioxidant and antimicrobial activities of Syrah grape skin residue but nevertheless it has bring us new important evidence by well conducted experiment. This scientific paper is well written so I recommend that this paper be published after all the proposed corrections, which I have cited to authors. This is a high quality article and I said that needs minor revision because of the few errors that need to be corrected. After correction is made, the manuscript can be published.

Author Response

(The authors gave the same response as above.)

Round 2

Reviewer 1 Report

Dear Authors, thank you for providing such detailed responses to my comments. In my opinion the manuscript has been significantly improved and I recommend its publication in Biology journal.

I have only one minor comment for lines 173-175 – for me DMSO is a negative control (as it should have no effect on the viability of bacteria) and amphotericin B is a positive control (as it has toxic effect for tested bacteria).

Author Response

– Please make revision and submit according to the suggestions of academic editors: Consider in lines 173-175 – for me DMSO is a negative control (as it should have no effect on the viability of bacteria) and amphotericin B is a positive control (as it has toxic effect for tested bacteria).

Answer: The point was revised, as requested by academic editors.
